# *Thymus* Species from Romanian Spontaneous Flora as Promising Source of Phenolic Secondary Metabolites with Health-Related Benefits

**DOI:** 10.3390/antiox12020390

**Published:** 2023-02-06

**Authors:** Mihai Babotă, Oleg Frumuzachi, Alexandru Nicolescu, Maria Inês Dias, José Pinela, Lillian Barros, Mikel Añibarro-Ortega, Dejan Stojković, Tamara Carević, Andrei Mocan, Víctor López, Gianina Crișan

**Affiliations:** 1Department of Pharmaceutical Botany, “Iuliu Hațieganu” University of Medicine and Pharmacy, Gheorghe Marinescu Street 23, 400337 Cluj-Napoca, Romania; 2Laboratory of Chromatography, Institute of Advanced Horticulture Research of Transylvania, University of Agricultural Sciences and Veterinary Medicine, 400372 Cluj-Napoca, Romania; 3Centro de Investigação de Montanha (CIMO), Instituto Politécnico de Bragança, Campus de Santa Apolónia, 5300-253 Bragança, Portugal; 4Laboratório Associado para a Sustentabilidade e Tecnologia em Regiões de Montanha (SusTEC), Instituto Politécnico de Bragança, Campus de Santa Apolónia, 5300-253 Bragança, Portugal; 5Department of Plant Physiology, Institute for Biological Research “Siniša Stanković”—National Institute of Republic of Serbia, University of Belgrade, Bulevar despota Stefana 142, 11000 Belgrade, Serbia; 6Instituto Agroalimentario de Aragón, IA2, Universidad de Zaragoza-CITA, 50830 Zaragoza, Spain; 7Department of Pharmacy, Faculty of Health Sciences, Universidad San Jorge, Villanueva de Gállego, 50830 Zaragoza, Spain

**Keywords:** *Thymus* sp., wild thyme, antioxidant activity, enzyme inhibition, antimicrobial activity

## Abstract

Wild thyme aerial parts (*Serpylli herba*) are recognized as a valuable herbal product with antioxidant, anti-inflammatory, and antibacterial effects. Although pharmacopoeial regulations allow its collection exclusively from *Thymus serpyllum*, substitution with other species is frequent in current practice. This study analyzed the phenolic composition, antioxidant, and enzyme-inhibitory and antimicrobial activity of the hydroethanolic extracts obtained from five Romanian wild thyme species (*Thymus alpestris*, *T. glabrescens*, *T. panonicus*, *T. pulcherimus* and *T. pulegioides*). The analysis of individual phenolic constituents was performed through LC-ESI-DAD/MS^2^, while for the in vitro evaluation of antioxidant potential, TEAC, FRAP, DPPH, TBARS and OxHLIA assays were employed. The anti-enzymatic potential was tested in vitro against tyrosinase, *α*-glucosidase and acetylcholinesterase. High rosmarinic acid contents were quantified in all species (20.06 ± 0.32–80.49 ± 0.001 mg/g dry extract); phenolic acids derivatives (including salvianolic acids) were confirmed as the principal metabolites of *T. alpestris* and *T. glabrescens*, while eriodictyol-*O*-di-hexoside was found exclusively in *T. alpestris*. All species showed strong antioxidant potential and moderate anti-enzymatic effect against *α*-glucosidase and acetylcholinesterase, showing no anti-tyrosinase activity. This is the first detailed report on the chemical and biological profile of *T. alpestris* collected from Romanian spontaneous flora.

## 1. Introduction

The *Thymus* species belong to the most representative plants of the Lamiaceae family, widely used both for their food and medicinal applications. This genus comprises an enormous number of species, sub-species and varieties, which are described mainly as spontaneous shrubs or herbs [1,2,3]. Although the origin of this genus is linked to the Mediterranean regions, thyme species are also widespread in temperate areas, including Romania [4,5]. According to the latest report of Sârbu et al. [6], 16 wild thyme species could be found in Romanian spontaneous flora, excluding their inferior taxonomic sub-divisions (sub-species, varieties).

The health-promoting benefits of *Thymus* species are well known and have been recognized for thousands of years, being supported through traditional use, as well as based on pharmacologically proven evidence. Moreover, competent authorities (i.e., WHO, EMA, FDA) issued monographs for thyme species officially recognized as medicinal plants, including wild-type ones [7,8]. According to European Pharmacopoeia, the herbal product *Serpylli herba* (defined as the aerial parts of wild thyme *T. serpyllum* L.) could be collected only from *T. serpyllum* L., but in current practice, substitution with other species is frequent due to low availability of the officinal one in different areas and its similar botanic features to other related *Thymus* species [9]. In Romania, wild thyme is known under the common name “*cimbrișor*”, an herbal product available on the market collected from various spontaneous *Thymus* species. The chemical composition of these species has been intensively studied; the main phytoconstituents responsible for their bioactive potentials are volatile oil and non-volatile fraction (rich in various phytochemicals, including phenolic acids, flavonoids, non-volatile terpenes, tannins or anthocyanins) [1,10,11,12,13].

Among the *Thymus* ssp. volatile constituents, thymol and carvacrol are the most known and studied molecules with well-established applications in human health and food industries [14], as well as modulators for biotechnological purposes (i.e., the improvement of structural and functional properties of proteins from microalgae Spirulina and Chlorella) [15]. On the other hand, considering the complexity of non-volatile metabolites’ profile (mainly polyphenol-type compounds), the interest in these species is still increasing due to the presence of several chemical constituents intensively studied at the moment for their promising bioactive potentials. As with other Lamiaceae family members, *Thymus* species were proven as being rich in phenolic acids, particularly rosmarinic acid [16,17]. Additionally, the latest research works have paid special attention to salvianolic acids, another sub-class of phenolic acids derivatives, which occurs in the aerial parts of different thyme species [16,17,18,19]. The distribution of the above-mentioned secondary metabolites is strongly correlated with the botanical identity of the originating species, as well as with the environmental or processing conditions applied to the plant material pre- or post-harvesting [19,20].

The actual concerns regarding thyme species are also focused on the methods employed in the processing and quality control, which exert an important impact against the bioactive potential of the phyto-preparations obtained from this herbal drug. A wide range of thyme-based products can be found on the market (herbal teas, liquid/dry extracts, syrups, tablets, capsules, etc.), mainly registered as dietary supplements [16]. Their production requires the use of different extraction procedures, aiming to assure a maximum recovery of bioactive compounds from the plant material and minimizing the use of solvents, extraction time and environmental impact. Even though modern extraction techniques (i.e., microwave/ultrasound/enzyme-assisted extraction, deep eutectic solvents extraction) are intensively promoted as suitable alternatives, which can meet the aforementioned criteria [21,22,23], the classic extraction methods (i.e., maceration, percolation) are still intensively applied in order to obtain thyme extracts, mostly because they are described as simple procedures and do not require the use of complex equipment. Additionally, their use is supported through long-term traditional use, pharmacopoeial regulations or other officially recognized monographs [8,24].

From the analytical perspective, chromatographic techniques are frequently cited as being suitable for the quality control of herbal drugs and phytomedicines originating from the *Thymus* species. Qualitative and quantitative distribution of principal metabolites is used in current practice as the main criteria for quality control, including the authentication of plant material or standardization. These goals adopt the use of complementary, fast, reproducible and robust methods, which can be modified for current use and are able to assure both proper identification and quantification of the marker compounds. For the assessment of volatile compounds, gas chromatography coupled with mass spectrometry (i.e., GC-MS, GC-MS/MS, GC-HRMS) [25,26,27] is one of the most used methods; non-volatile components of different thyme species are frequently analyzed through liquid chromatography (HPLC, UPLC) coupled with various detection methods (DAD, UV-Vis, MS) [19,20,28]. LC–MS is recognized as one of the best analytical techniques used for routine characterization of the phenolic profile of complex matrices, including herbal products, offering multiple advantages, such as short time of analysis, the requirement for minimum pre-treatment of the samples and direct detection without mandatory derivatization steps [29].

Considering the intensive use of wild thyme as herbal tea and the various *Thymus* species found in Romanian spontaneous flora, which can serve as a source for this product, our present study intended to comparatively evaluate the chemical and bioactive profile of five common wild thyme species, namely *Thymus alpestris*, *T. glabrescens* ssp. *glabrescens*, *T. panonicus* ssp. *auctus*, *T. pulcherimus* and *T. pulegioides* ssp. *pulegioides*. At the same time, this work intended to present, for the first time, an in-depth overview on the phenolic composition and antibacterial potential of *T. alpestris*, a less studied wild thyme species.

## 2. Materials and Methods

### 2.1. Standards and Reagents

For total phenolic content measurement (TPC), the following reagents and standards were used: Folin–Ciocalteu reagent (F9252), carbonic acid disodium salt (Na_2_CO_3_) (223530) and gallic acid (G7384). In the assessment of TFC, aluminum chloride (AlCl_3_) and rutin (PHL89270) were used. Antioxidant capacity assays required the following reagents and standards: ABTS (A1888), potassium persulfate (K_2_S_2_O_8_), DPPH (D9132), sodium acetate (CH_3_COONa), acetic acid (CH_3_COOH), hydrochloric acid (HCl), ferric chloride hexahydrate (FeCl_3_ × 6H_2_O), 2,4,6-tris(2-pyridyl)-S-triazine (TPTZ) (T1253), xanthine (X7375), sodium carbonate (Na_2_CO_3_), nitro blue tetrazolium chloride (NBT) (N6876), xanthine oxidase (X1875) and 6-hydroxy-2,5,7,8-tetramethylchroman-2-carboxylic acid (Trolox) (238813). Furthermore, in order to assess the enzyme-inhibitory capacity of the samples, the following reagents and standards were used: *α*-D-glucoside glucohydrolase from *Saccharomyces cerevisiae* (G5003), *Electrophorus electricus* cholinesterase (C3389), acarbose (A8980), galantamine (Y0001279), dimethyl sulfoxide (DMSO), 5,5′-dithiobis(2-nitrobenzoic acid) (DTNB) (D8130), 4-nitrophenyl-*β*-D-glucopyranoside (*p*-NPG) (N7006), 2-amino-2-(hydroxymethyl)-1,3-propanediol (Tris base) (TRIS-RO). Where a product number is specified, all the reagents and standards were purchased from Sigma-Aldrich Chemie GmbH (Taufkirchen, Germany).

For the LC–MS analysis, standard compounds (apigenin 6-*C*-glucoside ≥ 99%, apigenin 7-*O*-glucoside ≥ 99%, caffeic acid ≥ 99%, chlorogenic acid ≥ 99%, *p*-coumaric acid ≥ 90%; naringenin ≥ 99%, quercetin 3-*O*-glucoside ≥ 99%, rosmarinic acid ≥ 99% HPLC purity) were supplied by Extrasynthèse, Genay, France; all solvents used were HPLC grade.

### 2.2. Plant Material

For the purpose of obtaining proper extracts of the plants included in this study, the aerial parts of *T. alpestris*, *T. glabrescens*, *T. pannonicus*, *T. pulcherimus* and *T. pulegioides* were collected during the flowering period from different areas of Romania (see Table 1). Afterward, in order to achieve a suitable form of processing, plant material was sorted, subjected to an authentication procedure and dried at room temperature in a dark place until constant mass was attained. After the plant material reached constant mass, it was kept in a separate paper bag in the herbarium of the Pharmaceutical Botany Department of Iuliu Hațieganu University of Medicine and Pharmacy, Cluj-Napoca, Romania, until the extraction procedure was implemented.

### 2.3. Extraction Procedure

To assure a maximum extraction yield of bioactive compounds from the studied plants, dried plant material was powdered using a laboratory mill (Grindomix^®^ GM 200, Retsch Gmbh., Haan, Germany) and manually sieved (1 mm standard sieve according to *Ph.Eur. 10.8*) to ensure uniformity of the particles. According to the pharmacopoeial requirements (*Ph.Eur. 10.8*, *Romanian Pharmacopoeia 10th ed*), bioactive compounds from the studied plants were extracted using ethanol 70% as solvent (1:10 *w/v* powdered herb:solvent ratio); this extraction procedure aimed to simulate the traditional preparation method for tinctures (maceration in a dark place for 10 days at room temperature) [30,31]. 

At the end of the maceration time, the extraction mixtures were filtered under reduced pressure using paper filters (grade 202, retention 5–8 μm, diameter 110 μm, Frisenette ApS, Knebel, Denmark). Complete evaporation of ethanol from the extraction liquors was achieved using a rotary evaporator. Afterward, the extracts were lyophilized (Biobase^®^ BK-FD18S, Biobase group, Jinan, Shandong, China) and placed in a desiccator until further steps. At the end of the extraction procedure, 5 herbal preparations were prepared for future analyses, namely: **T. alp** (*T. alpestris* dry extract), **T. glb** (*T. glabrescens* dry extract), **T. pan** (*T. pannonicus* dry extract), **T. plc** (*T. pulcherimus* dry extract), **T. plg** (*T. pulegioides* dry extract).

### 2.4. Chromatographic Profiling of the Extracts

Each sample (10 mg) was dissolved using 2 mL of hydromethanolic solvent (20:80, *v/v*) and filtered through 0.22 μm membrane filter. Further, the extracts were analyzed using a Dionex Ultimate 3000 UPLC (Thermo Scientific, San Jose, CA, USA) system equipped with a diode array detector coupled to an electrospray ionization mass detector (LC-DAD-ESI/MS^n^) [32]. Online detection was achieved using a Diode Array Detector DAD (280, 330 and 370 nm as preferential wavelengths) coupled with an ESI mass spectrometer working in negative mode (Linear Ion Trap LTQ XL mass spectrometer, Thermo Finnigan, San Jose, CA, USA).

Chromatographic separation was achieved with a Waters Spherisorb S3 ODS-2C18 (3 m, 4.6 mm × 150 mm, Waters, Milford, MA, USA) column thermostatized at 35 °C. The solvents used were: (A) 0.1% formic acid in water, (B) acetonitrile. The elution gradient established was isocratic 15% B (5 min), 15% B to 20% B (5 min), 20–25% B (10 min), 25–35% B (10 min), 35–50% B (10 min), and re-equilibration of the column was performed using a flow rate of 0.5 mL/min. The injection volume was 100 µL. 

MS detection was performed in negative mode. Nitrogen served as the sheath gas (50 psi); the system was operated with a spray voltage of 5 kV, a source temperature of 325 °C, a capillary voltage of −20 V. The tube lens offset was kept at a voltage of −66 V. The full scan covered the mass range from *m/z* 100 to 2000, with a mass accuracy of 0.15 Da, a peak width of 07 FWHM and a scan rate of 16,667 Da/sec. The collision energy used was 35 (arbitrary units). Data acquisition was carried out with Xcalibur^®^ data system (ThermoFinnigan, San Jose, CA, USA).

The identification of phenolic compounds was performed by comparing their retention times, UV–vis and mass spectra with those obtained from standard compounds; otherwise, compounds were tentatively identified by comparing the obtained information with the available data reported in the literature. For quantitative evaluation, a calibration curve for each available phenolic standard was constructed based on the UV signal; for the identified phenolic compounds for which a commercial standard was not available, quantification was performed through the calibration curve of the most similar available standard, and the results were expressed as mg/g of the extract [32,33,34].

### 2.5. Evaluation of Total Phenolic Content (TPC) of Samples

TPC was measured based on a protocol previously published by Tanase et al. [35]. In brief, 10 mL of Folin–Ciocalteu phenol reagent was diluted to a final concentration of 10%, using distilled water. Afterward, 100 μL of 10% Folin–Ciocalteu phenol reagent and 20 μL of the extract (re-dissolved in 70% ethanol solution to a final concentration of 1 mg/mL) were pipetted on a microplate and pre-incubated (in dark, 3 min, room temperature). Further, to ensure an alkaline medium, the mixture was completed with 80 μL of 7.5% Na_2_CO_3_ solution and incubated for another 30 min in the same conditions. Finally, after 30 min incubation time, the plate was read at 760 nm. The results were calculated as milligrams of gallic acid equivalents (GAE)/g of lyophilized extract.

### 2.6. Evaluation of Total Flavonoid Content (TFC) of Samples

A protocol described by Tanase et al. [35] was used for the TFC assay. Briefly, 2% AlCl_3_ solution (100 μL) was pipetted together with the extract solution (100 μL) (re-dissolved in 70% ethanol solution to a final concentration of 1 mg/mL) on a microplate and incubated in the same conditions as described in the aforementioned protocol. After 10 min of incubation, the extinction was measured at 420 nm. The final results were calculated as milligrams of rutin equivalents (RE)/g of lyophilized extract.

### 2.7. Total Antioxidant Capacity

#### 2.7.1. DPPH Radical-Scavenging Activity Assay

An elaborate characterization of the implemented protocol was described by Babotă et al. [36]. In brief, 270 μL of 0.004% DPPH methanol solution and 30 μL of the extract (re-dissolved in 70% methanol solution to a final concentration of 1 mg/mL) were pipetted on a microplate and incubated for 30 min in a dark place and at room temperature. After the incubation had occurred, the plate was read at 517 nm, and the final results were calculated as milligrams of Trolox equivalents (TE)/g of lyophilized extract.

#### 2.7.2. Ferric Reducing Antioxidant Power (FRAP) Assay

The first step in implementing this previously reported assay [36] involved the preparation of 10 mL of acetate buffer (pH = 3.6, 100 mM). Afterward, to generate a FRAP reagent solution, 10 mL of acetate buffer were mixed with 1 mL of HCl solution (40 mM) and 1 mL of FeCl_3_ solution (20 mM). To evaluate the ferric reducing antioxidant power of the studied plants, 175 μL of FRAP were added to 25 μL of the extract (re-dissolved in 70% ethanol solution to a final concentration of 0.25 mg/mL) and incubated under previously mentioned conditions; the reaction was spectrophotometrically monitored at 593 nm. The results of the FRAP assay were expressed as milligrams of Trolox equivalents (TE)/g of lyophilized extract.

#### 2.7.3. Trolox Equivalent Antioxidant Capacity (TEAC) Assay

The first step in implementing this previously reported assay [36] involved the generation of the TEAC radical stock solution. Therefore, 10 mL of ABTS^+^ radical solution (2.15 mM) were mixed with K_2_S_2_O_8_ solution (10 mL, 1.40 mM), and the mixture was incubated for ~24 h in a dark place and at room temperature. Subsequently, the radical stock solution was diluted with distilled water to a final absorbance of 0.700 ± 5% at 734 nm. Afterward, 200 μL of radical stock solution and 20 μL of sample solution (re-dissolved in 70% ethanol solution to a final concentration of 1 mg/mL) were mixed together and incubated for 6 min at room temperature, in a dark place. At the end of the incubation time, the absorbance was monitored at 734 nm. The final results were calculated as milligrams of Trolox equivalents (TE)/g of lyophilized extract.

#### 2.7.4. Superoxide Radical-Scavenging Activity Assay

To evaluate the capacity of the studied plants to neutralize superoxide radical (O_2_^•−^), the xanthine/xanthine oxidase method [37] was used. Hence, 240 μL of a mixture consisting of xanthine, NBT and Na_2_CO_3_ were added to 30 μL of the extract (re-dissolved in phosphate buffer, pH = 7.4, 100 mM, to obtain different dilutions between 0.975 and 1000 μg/mL) and 30 μL of 0.16 U/mL xanthine oxidase. Incubation at 37 °C for 5 min was required, and the absorbance of the reaction mixture was read spectrophotometrically at 560 nm. The ability of plant extracts to scavenge O_2_^•−^ radicals was evaluated by measuring the intensity of the reaction color, which represented the transformation of NBT to the blue chromogen formazan. Low-intensity reaction color indicated by low absorbance represented increased superoxide anion scavenging activity of the extracts. Gallic acid was used as a reference substance. The results were expressed as radical-scavenging capacity (RSC) calculated as follows (Equation (1)):(1)$$RSC\;\left(\%\right)=\left(Abs_{control}-Abs_{sample}\right)/Abs_{control}\times 100$$

The *RSC* for each concentration was calculated and plotted on a graph, and the final results were expressed as IC_50_.

#### 2.7.5. Thiobarbituric Acid Reactive Substances (TBARS) Formation Inhibition Capacity Assay

The TBARS formation inhibition capacity assay was implemented following a protocol described by Babotă et al [23]. In the first phase of the assay, FeSO_4_ (0.01 mM), L-ascorbic acid (0.1 mM) and the sample were mixed in equal parts (100 μL) in a 2 mL Eppendorf reaction tube. The mixture was subjected to incubation (60 min, 37 °C) and treated with trichloroacetic acid (28% *w/v*, 500 μL) and thiobarbituric acid (TBA, 2% *w/v*, 380 μL). The formation of the malondialdehyde (MDA)–TBA complex was initiated by heating the tubes (20 min, 80 °C), its quantification being measured at 532 nm in the supernatants previously separated through centrifugation (3000 rpm, 10 min). The results were expressed as IC_50_ values (μg/mL).

#### 2.7.6. Oxidative Hemolysis Inhibition Assay (OxHLIA)

The protocol used for the implementation of this assay was described by Añibarro-Ortega et al. [38]. Briefly, 200 µL of 2.8% erythrocyte solution (*v/v*) were mixed with 400 µL of the sample solution dissolved in PBS. After 10 min of incubation at 37 °C, 200 μL of 2,2′-azobis(2-methylpropionamidine) dihydrochloride (160 mM) were added, and the optical density was kinetically measured at 690 nm on an ELx800 microplate reader (Bio-Tek Instruments, Winooski, VT, USA) until complete hemolysis. Trolox was used as positive control (different concentrations ranging from 7.81 to 125 μg/mL); PBS was used as negative control, and distilled water was used as baseline. The results were calculated as IC_50_ values (µg/mL) for a Δ*t* of 60 min.

### 2.8. α-Glucosidase Inhibition Assay

The protocol used to evaluate the potential of the studied plants to inhibit *α*-glucosidase was described by Babotă et al. [36]. Hence, 50 μL of the extract (re-dissolved in 100 mM PBS, pH = 6.8, at concentrations between 46.87 and 3000 μg/mL) were pipetted together with 50 μL of *α*-glucosidase enzyme solution (750 U/L) and 50 μL of PBS and incubated for 10 min at 37 °C in a dark place. In the next step, the mixture was completed with *p*-NPG (50 μL) and re-incubated for 10 min. Extinction was spectrophotometrically monitored at 405 nm. IC_50_ values (μg/mL) were calculated by reporting acarbose as a reference substance.

### 2.9. Acetylcholinesterase (AChE) Inhibition Assay

The AChE inhibition assay was implemented using a protocol described by Tanase et al. [35]. In the first step, the reaction mixture containing Tris-HCl buffer (50 μL, 50 mM, pH = 8), the extract (re-dissolved in Tris-HCl buffer to obtain dilutions between 500 and 10,000 μg/mL), DTNB (0.9 mM, 125 μL) and the AChE enzyme (25 μL, 78 U·L^−1^) was subjected to incubation (15 min, 37 °C) in order to let the sample solution exert its inhibition potential. At the end of pre-incubation time, the mixture was completed by adding 25 μL of ATCI (4.5 mM) and re-incubated under the same conditions. Finally, the absorbance was measured spectrophotometrically (405 nm). Galantamine was used as a reference substance. The results were expressed as IC_50_ values (μg/mL).

### 2.10. Assessment of the Antibacterial Effects

The potential of the studied plants to inhibit antimicrobial growth was assessed based on a protocol described by Babotă et al. [36]. The parameters that described microbial growth inhibition were minimum inhibitory concentrations (MICs), and the parameters that described bactericidal and fungicidal potentials were minimum bactericidal concentrations (MBCs) and minimum fungicidal concentrations (MFCs), respectively. The extracts obtained from the *Thymus* species were tested for their antibacterial potential against Gram-positive and Gram-negative bacteria, namely: *Staphylococcus aureus* (ATCC 11632), *Bacillus cereus* (clinical isolate), *Listeria monocytogenes* (NCTC 7973), *Escherichia coli* (ATCC 25922), *Salmonella enterica* subsp. *enterica* serovar Typhimurium (ATCC 13311) and *Enterobacter cloacae* (ATCC 35030). Furthermore, six micromycetes were used for the evaluation of the antifungal capacity of the studied plants, namely: *Aspergillus fumigatus* (human isolate), *Aspergillus niger* (ATCC 6275), *Aspergillus versicolor* (ATCC 11730), *Penicillium funiculosum* (ATCC 36839), *Trichoderma harzianum* (TH-IS005-12) and *Penicillium verrucosum var. cyclopium* (food isolate). The microbial strains used were obtained from the Mycological Laboratory, Department of Plant Physiology, Institute for Biological Research “Siniša Stanković”, University of Belgrade, Serbia. The results were expressed as mg/mL. 

### 2.11. Statistical Analysis

The results were expressed as the mean ± standard error of experiments performed in triplicate. Statistical analysis of the data was performed using GraphPad Prism 9 software. One-way analysis of variance (ANOVA) followed by Tukey’s HSD were used in order to assess the significant differences (*p* < 0.05) between species.

## 3. Results

### 3.1. TPC and TFC Assessment

Considering the importance of phytocompounds in health and disease prevention, as well as the multilateral implication of total phenolic and flavonoid content of different plant sources, we aimed to evaluate these parameters for our species. After conducting appropriate assays to evaluate the contents of these compounds, the results are presented in Figure 1.

Among the *Thymus* species studied, the highest average concentration of TPC was evaluated for the T. glb extract (150.01 mg/g extract), while the lowest average concentration was evaluated for the T. plg extract (132.08 mg GAE/g extract). The average ascending order of TPC in extracts of the studied species of *Thymus* genera was as follows: T. plg < T. plc < T. alp < T. pan < T. glb. However, no significant differences were observed among the *Thymus* species included in the study with regard to their total phenolic content.

With reference to TFC of the *Thymus* species studied, the highest average concentration of flavonoids was evaluated for the T. plc extract (55.48 mg RU/g extract), while the lowest average concentration was evaluated for the T. pan extract (34.51 mg RE/g extract). The average ascending order of TFC in extracts of the studied species of *Thymus* genera was as follows: T. pan < T. glb < T. plg < T. alp < T. plc. There were observed significant differences among all the studied species (Figure 1).

### 3.2. Individual Phenolic Profile

The method employed for the evaluation of individual phenolic constituents of the analyzed *Thymus* extracts confirmed the presence of 13 compounds (Table 2). The first eluted compound showed *λ*_max_ at 285 nm, and released after fragmentation the fragment with *m/z* 287 ([eriodictyol–H]^−^) after consecutive loss of two hexosyl moieties (−162 mu); hence, its identity was attributed to eriodictyol-*O*-di-hexoside [39]. The flavonoids group comprised kaempferol (3), quercetin (6) and luteolin (7, 8) derivatives, identified as -*O*-hexosides (loss of hexosyl moiety, −162 mu) or hexuronides. Phenolic acids derivatives (compounds 4, 5, 9–13) reached the highest amounts in all the extracts tested. Rosmarinic acid ([M–H]^−^ at *m/z* 359) was identified through comparison with the available standard, while salvianolic acids (salvianolic acid A and B) were recognized based on their ability to lose a fragment with *m/z* 198 after MS^2^ collision, corresponding to the danshensu moiety (common monomeric constituent found in the structure of all salvianolic acids) [40]. As could be observed, except for their retention times, compounds 10 and 11 showed identical chromatographic features; hence, they were assumed as the isomers of salvianolic acid A ([M–H]^−^ at *m/z* 493) [39].

Regarding the quantitative distribution of compounds analyzed in the *Thymus* species extracts, several differences were observed. Eriodictyol-*O*-di-hexoside and kaempferol-*O*-dihexoside were found exclusively in the *T. alpestris* extract, while salvianolic acid B, salvianolic acid I and its isomer were quantified only in the T. alp and T. pan samples. The dominance of several phenolic classes was also observed for the analyzed species, allowing us to divide them in two distinct groups. The first group (comprising *T. alpestris* and *T. glabrescens* ssp. *glabrescens*) was characterized as possessing higher contents of flavonoid-type metabolites, while the second group (*T. pulegioides* ssp. *pulegioides*, *T. panonicus* ssp. *auctus*, *T. pulcherimus*) possessed higher contents of phenolic acids derivatives. These findings are in line with previous reports on the chemical profile of several *Thymus* species from Romanian spontaneous flora, including the above-mentioned ones [9,23,41,42].

### 3.3. Antioxidant Potential of Selected Extracts

In order to present a comprehensive evaluation of the total antioxidant capacity of the *Thymus* species studied, six antioxidant capacity assays were implemented in this study. The consideration underlying this decision was related to the broad variability existing in between and among antioxidant capacity assays. By evaluating each species through all six assays, we wanted to decrease the existing variability and to increase the degree of completeness of the evaluation performed on the total antioxidant capacity. Data provided after the evaluation of antioxidant activity through DPPH, FRAP, TEAC, OxHLIA and TBARS methods are presented in Figure 2.

DPPH radical-scavenging activity assay is a simple and rapid method, inexpensive and widely used to characterize the ability of plant extract compounds to act as hydrogen donors or free radical scavengers [43]. The highest capacity to scavenge DPPH radical was observed for the extract obtained from *T. pannonicus* (80.71 mg TE/g extract), while the lowest capacity was observed for the extract obtained from *T. glabrescens* (78.99 mg TE/g extract). Although the average DPPH radical-scavenging capacity of the extracts was 80.16 ± 1% mg TE/g extract, a significant difference regarding the radical-scavenging capacity of the extracts was observed between T. alp and T. glb, T. glb and T. pan, and T. glb and T. plc. 

The ferric reducing antioxidant power (FRAP) assay is a simple test to measure the capacity of plant extracts to reduce ferric to ferrous ion at acidic pH [44]. Analogous to DPPH radical-scavenging activity assay, the highest capacity to reduce ferric to ferrous ion was observed for the T. pan extract (521.34 mg TE/g extract), while the lowest capacity was observed for the T. plc extract (431.73 mg TE/g extract). A significant difference was observed for the result between the *T. pulcherimus* extract and all three extracts of *T. alpestris*, *T. glabrescens* and *T. pannonicus*, while no significant difference was observed between any other species.

The Trolox equivalent antioxidant capacity (TEAC) assay is based on the ability of plant extracts to scavenge ABTS^+^ radical [45]. Even though the TEAC assay is a rapid method when implemented, it requires preliminary preparation of the radical solution at least 16 h before the determination itself, which might be an important limitation regarding its application. Notwithstanding this, the tested extracts showed a high ability to neutralize ABTS^+^ radical, with the highest ability exerted by the T. pan extract (152.32 mg TE/g extract), while the lowest ability was demonstrated by the T. plc extract (148.79 mg TE/g extract). A significant difference regarding the results obtained in this test was observed only between the extracts mentioned earlier, the T. pan and T. plc extracts, respectively, while a significant difference between the other extracts was absent. 

Oxidative hemolysis inhibition (OxHLIA), a simple cell-based assay, evaluates the inhibition of free radical damage exerted on red blood cell membranes by antioxidants. The main advantage of this assay is related to its biological relevance [46]. The highest potential to inhibit erythrocyte hemolysis was observed for the T. alp extract (1.79 mg/mL), followed by the T. plg extract (2.86 mg/mL), while the lowest erythrocyte hemolysis capacity was observed for the T. plc extract (6.91 mg/mL). A statistically significant difference was observed between all tested extracts (except for the T. glb and T. pan extracts). Moreover, a statistically significant difference was observed between the extracts and the standard compound used (Trolox, 21.80 mg/mL).

Thiobarbituric acid reactive substances (TBARS) assay measures lipid peroxidation in biological fluids. Like the OxHLIA assay, TBARS is a good indicator of the levels of oxidative stress within a biological sample [47]. In our analysis, the highest potential to inhibit lipid peroxidation was observed for the T. pan extract (6.29 mg/mL), followed by the T. glb extract (6.80 mg/mL), while the lowest lipid peroxidation capacity was observed for the T. plg extract (17.36 mg/mL). A statistically significant difference was observed between all the extracts tested and between the extracts tested and the standard compound used (Trolox, 22.69 mg/mL).

Superoxide radical (O_2_^•−^) can induce different levels of damage, which can impair critical biomacromolecules and alter different biological processes, such as gene expression. Impaired gene expression is a causative factor for mitogenesis, mutagenesis and cell death [48]. In order to evaluate the capacity of the studied extracts to scavenge superoxide radicals, O_2_^•−^ radicals were generated by the xanthine/xanthine oxidase system. The results of the assay are presented in Figure 3.

The *Thymus* extracts showed a high potential to neutralize O_2_^•−^ radicals. The IC_50_ values of the extract were situated in a range between 0.91 and 1.21 μg/mL, with no statistical differences between the extracts. The IC_50_ for gallic acid was 0.046 μg/mL.

### 3.4. Enzyme Inhibition Assays

*α*-Glucosidase is an enzyme with important implications for carbohydrates digestion, and its inhibition plays a critical role in therapeutic management of type 2 diabetes [49]. Acetylcholinesterase hydrolyzes the neurotransmitter acetylcholine into choline and acetate, being involved in the development of neurodegenerative conditions, including Alzheimer’s disease [50]. The results of the inhibitory potential of *Thymus* species to inhibit these two enzymes are presented in Figure 4.

All the extracts showed a high-to-moderate potency against *α*-glucosidase enzyme. The highest capacity to inhibit *α*-glucosidase was shown by the T. glb extract (IC_50_ = 296.82 μg/mL), followed by the T. alp (IC_50_ = 366.14 μg/mL) and T. pan (IC_50_ = 388.33 μg/mL) extracts. A statistical difference was observed between the T. alp, T. glb, T. pan extracts and T. plc, T plg extracts. Furthermore, with regard to significance, the T. alp, T. glb and T. pan extracts exerted the same enzyme inhibition capacity as standard acarbose compound (IC_50_ = 382.18 μg/mL).

Contrarily, only T. alp, T. glb and T. plg demonstrated a capacity to inhibit AChE; however, the results are very weak compared to standard galantamine compound (IC_50_ = 3.37 μg/mL). With regard to inter-species activity, the T. glb (IC_50_ = 2006.32 μg/mL) extract was the most potent one, followed by the T. alp (IC_50_ = 4406.38 μg/mL) and T. plg (IC_50_ = 4962.09 μg/mL) extracts. The IC_50_ values of T. alp and T. plg were statistically insignificant, while all the other values were statistically significant.

Detailed results regarding in vitro superoxide radical-scavenging activity and enzyme-inhibitory potential of *Thymus* species and reference substances are summarized in Table 3. 

### 3.5. Effects Exerted against Bacterial and Fungal Growth

These effects were evaluated by the measurement of MICs and MBCs using the microdilution method. As shown in Table 4, all micro-organisms tested showed the same sensitivity after exposure to the T. alp and T. glb extracts. Slight differences were observed in terms of MICs and MBCs values between Gram-positive and Gram-negative groups, the first ones (*S. aureus*, *B. cereus* and *L. monocytogenes*) exerting a lower response to T. pan, T. plc and T. plg samples. Overall, the antibacterial effect of the tested samples was weak in comparison with the positive controls used.

Conversely, the analyzed *Thymus* samples acted as moderate antifungal agents (Table 5), showing comparable MICs and MBCs values with those obtained for positive controls (a mean of 30–50% from the activity of ketoconazole and bifonazole). 

## 4. Discussion

The aerial parts of *Thymus* species are widely used for different purposes, including food seasoning or additive (both for flavoring and preserving properties), as well as for obtaining different herbal preparations with proven health benefits [2,12,51,52]. Therapeutic use of wild thyme is based on pharmacological evidence, pharmacopoeia recommendations and, mostly, on the data available from ethnopharmacology or traditional medicine. The actual trends in the research of these herbal products highlight their potential use as sources of plant-derived biomolecules with positive impact against different physiological and pathological processes.

In Romanian folk medicine, *Thymus* species from spontaneous flora belong to a group of the most well-known and used herbal medicines, with a long and well-documented tradition that supports therapeutic uses of these plants in human medicine. First of all, aqueous (i.e., decoctions, infusions) and hydroalcoholic herbal remedies obtained from different wild thyme species are widely recommended as remedies for respiratory tract ailments associated with inflammation, cough and infections [53,54,55]. All these uses are supported by results obtained after empirical herbal therapy, as well as by those obtained and validated through clinical research on human subjects [56,57,58,59]. The latest research opens new perspectives regarding the bioactive properties of spontaneous *Thymus* species and emphasizes the impact of their chemical composition variability on these properties.

As well as other plants from the Lamiaceae family, thyme species contain important amounts of volatile oils and non-volatile fraction; in the second category, the main part is covered by polyphenol-type compounds, especially phenolic acids and flavonoids. The distribution of these two types of metabolites is dependent on various factors, including botanical identity or the environment [20,60,61]. This statement was previously proved by Sarfaraz et al. [19] in a study focused on evaluating the molecular variations between eleven *Thymus* species, including the cultivated *T. vulgaris* and wild-type ones (i.e., *T. daensis*, *T. serpyllum*). The main compounds identified for all the species were rosmarinic acid, phenolic acids derivatives (caffeic, ferulic) and salvianolic acid derivatives; the same study reported an important flavonoid content (naringenin and epicatechin derivatives), especially for wild thyme species, suggesting the importance of these compounds as chemotaxonomic markers useful for distinction between species. Similar findings were reported by Boros et al. [61] regarding the chemical variances between *T. glabrescens* Willd., *T. pannonicus* All., *T. praecox* Opiz, *T. pulegioides* L. and *T. serpyllum* L. collected from Hungary and Romania; the LC–MS profile of the extracts obtained from the Romanian samples was characterized by the presence of apigenin and luteolin derivatives, as well as by higher contents of rosmarinic acid. As we mentioned already, our results emphasize a direct proportional relationship between the occurrence of flavonoids and phenolic acids derivatives in the studied species, which can be correlated with the effect of environmental factors. Hence, for the “mountain-type” wild thyme (*T. alpestris* and *T. pulegioides* ssp. pulegioides), flavonoids were found to be most abundant, while for the species originating from lower altitudes (*T. glabrescens* ssp. glabrescens, *T. panonicus* ssp. *auctus*, *T. pulcherimus*), phenolic acids derivatives seemed to be the main secondary metabolites.

The bioactive potential of the studied species was influenced by their chemical profile, especially for the anti-glucosidase and antimicrobial effects. The inhibition of *α*-glucosidase was higher for the extracts rich in flavonoids, which is in line with previous obtained results regarding the ability of these plant metabolites to act as potential antidiabetic agents [62,63]. As one of the therapeutic targets in diabetes mellitus is this enzyme, the research for novel *α*-glucosidase inhibitors has intensified in recent years, with several potent candidates being found among the flavonoids group. Tadera et al. [62] studied the anti-glucosidase properties of different isolated flavonoids, revealing elevated inhibition rates for the flavonol-type (i.e., quercetin—91%) and flavone-type (i.e., luteolin—92%) derivatives. Moreover, using natural flavonoids as scaffolds, Zhu et al. [63] obtained several compounds through chemical synthesis with augmented anti-glucosidase potential and improved binding activity to the active site of the enzyme. Of course, the antidiabetic potential of several thyme species (i.e., *T. spicata* var. *spicata*, *T. argaeus*, *T. praecox*) was also exploited in traditional medicine based on empirical ethnopharmacological observations [2,64]; for some of them, this activity has been clinically proven. Taleb et al. [65] demonstrated in a clinical trial that daily intake of *T. kotschyanus* infusion improves blood glucose levels and β-pancreatic-cell activity, while *T. fedtschenkoi* was proven as a potent inhibitor of α-glucosidase through intimate molecular mechanisms of several main constituents, which interact with the active sites of this enzyme [66]. Based on the observations of Mohammadi-Liri et al. [66], rosmarinic acid isolated from *T. fedtschenkoi* meets favorable structural features to establish interactions with ASP203, THR205 and GLN603 residues from the N-terminal region of the enzyme, having been found as the most potent chemical constituent of this species (IC_50_ 43.38 ± 0.05 μM). Recent reports also highlight salvianolic acids as novel potential α-glucosidase inhibitors. Molecular docking studies with mushroom glucosidase as the model proved the ability of salvianolic acid A to penetrate the active site of the enzyme, interact with key residues at this level, prevent the binding of the substrate and, finally, inhibit the catalytic action of the enzyme [67]. In a similar manner, Thang et al. [68] confirmed the inhibitory potential of salvianolic acid C against the same enzyme, this compound being able to establish hydrophobic interactions with the catalytic region in a similar manner as the substrate (*α*-D-glucose).

On the other hand, the mild-to-moderate antimicrobial activity exerted by the five species studied could be attributed to the synergistic effects of the main secondary metabolites found in the extracts. Even though *Thymus* species are recognized as effective antibacterial agents, mainly due to their volatile oils, several studies also confirmed bactericidal and/or bacteriostatic effects of the non-volatile fraction isolated from the aerial parts of wild thyme [69,70]. In a comparative study focused on evaluating the antimicrobial potency of *T. vulgaris*, *T. serpyllum*, *T. pulegioides* and *T. glabrescens* aerial parts, Varga et al. [70] found rosmarinic acid and several flavonoids (apigenin, dihydroquercetin, naringenin) as the main compounds responsible for the antibacterial effects of the hydromethanolic extracts obtained from these species. Similar results were reported by Arsenijević et al. [71,72] regarding the antibacterial potential of *T. pannonicus* herbal teas and hydromethanol extracts, correlated with the presence of rosmarinic acid and luteolin glycosides in their composition.

Hence, the present study emphasizes the important differences between the chemical and bioactive profile of several wild thyme species frequently collected from Romanian spontaneous flora as constituents of the herbal product *Serpylli herba*. Depending on the collection areas, the occurrence of different species in the final product could influence its quality, which requires the development of additional methods in order to assure the correct authentication of the herbal material and complete the data supplied through macro- and microscopic examination (based on the confirmation of botanical features). At the same time, to the best of our knowledge, this study offers the first detailed report on the phytochemical characterization of *T. alpestris*, a wild thyme species only reported in several regions of Europe (Austria, Liechtenstein, former Czechoslovakia, France, the Channel Islands and Monaco, Poland, Romania and Ukraine). As could be observed, from the chemical perspective, the extract obtained from *T. alpestris* seems to show a distinct chromatographic fingerprint in comparison with the other four species analyzed, being rich in flavonoid-type secondary metabolites. This pattern could explain its slightly augmented antioxidant and anti-glucosidase properties, as well as the ability to act as the strongest in vitro superoxide radical scavenger.

## 5. Conclusions

*Thymus alpestris*, *T. glabrescens* ssp. *glabrescens*, *T. panonicus* ssp. *auctus*, *T. pulcherimus* and *T. pulegioides* ssp. *pulegioides* collected from Romanian spontaneous flora were found to be phenolic-rich wild thyme species. Flavonoid-type secondary metabolites are more related to *T. alpestris* and *T. pulegioides* ssp. pulegioides extracts, while phenolic acid derivatives are more specific for *T. glabrescens* ssp. glabrescens, *T. panonicus* ssp. *auctus* and *T. pulcherimus*. The occurrence of the above-mentioned phytoconstituents is responsible for several bioactivities exerted by the extracts obtained from this species, including the antioxidant capacity, *α*-glucosidase inhibitory potential and mild-to-moderate antibacterial and antifungal potentials. Our findings support the use of the LC–MS method as a suitable tool in the analysis of the herbal drug *Serpylli herba*; the data provided through this technique serve as valuable criteria for the authentication and quality control of this product. Moreover, we emphasize the importance of the less studied *T. alpestris* as a valuable wild thyme species with potential applications in phytotherapy, our study presenting, for the first time, a detailed report on the chemical and bioactive features of this herb.

## Figures and Tables

**Figure 1 antioxidants-12-00390-f001:**
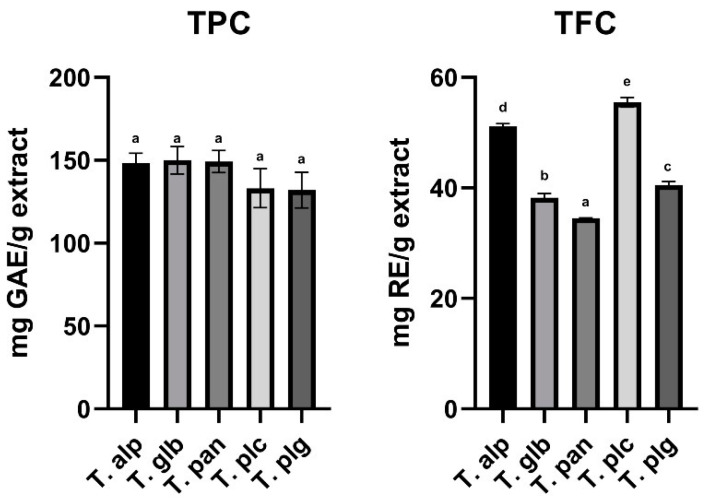
TPC and TFC values measured for the *Thymus* species studied. The results are expressed as average ± standard deviation of three parallel measurements. Statistical differences were assessed by one-way ANOVA, followed by Tukey’s HSD post hoc test (α = 0.05). Different lower case letters indicate significant differences between extracts. **T. alp**—*T. alpestris*, **T. glb**—*T. glabrescens*, **T. pan**—*T. pannonicus*, **T. plc**—*T. pulegioides*, **T. plg**—*T. pulcherimus*.

**Figure 2 antioxidants-12-00390-f002:**
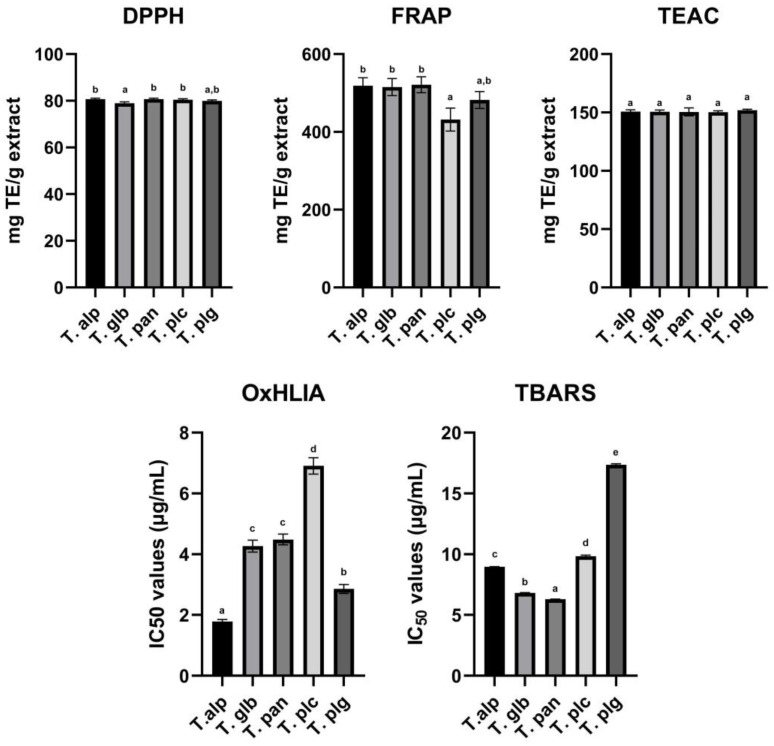
Results of the in vitro antioxidant capacity of *Thymus* species studied through DPPH, FRAP, TEAC, OxHLIA and TBARS assays. The results are expressed as average ± standard deviation of three parallel measurements. Statistical differences were assessed by one-way ANOVA, followed by Tukey’s HSD post hoc test (α = 0.05). Different lower-case letters indicate significant differences between extracts. **T. alp**—*T. alpestris*, **T. glb**—*T. glabrescens*, **T. pan**—*T. pannonicus*, **T. plc**—*T. pulegioides*, **T. plg**—*T. pulcherimus*.

**Figure 3 antioxidants-12-00390-f003:**
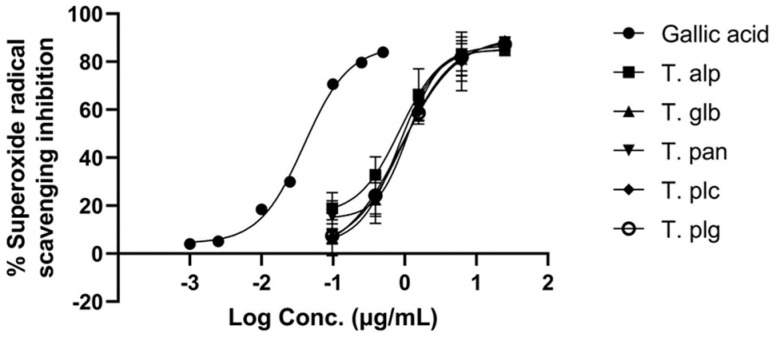
The concentration (expressed as logC, in μg/mL)-dependent antioxidant activity of *Thymus* extracts and gallic acid against superoxide radical generated by the xanthine/xanthine oxidase method. **T. alp**—*T. alpestris*, **T. glb**—*T. glabrescens*, **T. pan**—*T. pannonicus*, **T. plc**—*T. pulegioides*, **T. plg**—*T. pulcherimus*.

**Figure 4 antioxidants-12-00390-f004:**
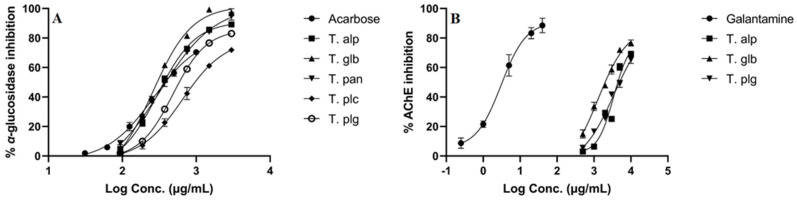
The concentration (expressed as logC, in μg/mL)-dependent enzyme-inhibitory activity of *Thymus* extracts against *α*-glucosidase (**A**) and acetylcholinesterase (AChE) (**B**). Acarbose and galantamine were used as reference substances in anti-*α*-glucosidase and anti-AChE assays, respectively. **T. alp**—*T. alpestris*, **T. glb**—*T. glabrescens*, **T. pan**—*T. pannonicus*, **T. plc**—*T. pulegioides*, **T. plg**—*T. pulcherimus*.

**Table 1 antioxidants-12-00390-t001:** Identification data for *Thymus* samples collected.

Species	Collection Date and Location
*T. alpestris*	July 2020, Rarău mountains, Suceava County
*T. glabrescens* ssp. *glabrescens*	June 2020, Ciucea, Cluj County
*T. pannonicus* spp. *auctus*	June 2018, Hida, Sălaj County
*T. pulcherrimus*	July 2018, Bucegi mountains, Brașov County
*T. pulegioides* spp. *pulegioides*	July 2020, Rarău mountains, Suceava County

**Table 2 antioxidants-12-00390-t002:** Identification and quantification data of phenolic compounds analyzed from *Thymus* samples. Statistical differences were evaluated by implementing one-way ANOVA, followed by Tukey’s HSD post hoc test (*p* < 0.05). Different lower-case letters indicate significant differences among *Thymus* species. nd—not detected.

Peak	Rt (min)	λ_max_ (nm)	[M-H]^−^ (*m/z*)	MS^2^ (*m/z*)	Tentative Identification	Content (mg/g Extract)				
						T. alp	T. glb	T. pan	T. plc	T. plg
**1**	5.98	285	611.2	449 (100), 287 (28)	Eriodictyol-*O*-di-hexoside	3.46 ± 0.003 ^a^	nd	nd	nd	nd
**2**	8.7	326	593.2	473 (100), 383 (19), 353 (25)	Apigenin-di-*C*-hexoside	nd	2.06 ± 0.006 ^b^	1.21 ± 0.001 ^a^	nd	nd
**3**	9.05	327	609.3	447 (100), 285	Kaempferol-*O*-dihexoside	7.69 ± 0.008 ^c^	nd	nd	0.6 ± 0.001 ^a^	0.81 ± 0.006 ^b^
**4**	12.7	285, 342	537.1	339 (100), 493 (21), 313	Salvianolic acid I	nd	7.17 ± 0.378 ^b^	6.69 ± 0.036 ^a^	nd	nd
**5**	13.4	343	537.1	339 (100), 493 (23)	Salvianolic acid I isomer	nd	13.11 ± 0.087 ^b^	9.18 ± 0.021 ^a^	nd	nd
**6**	14.6	342	477.2	301 (100)	Quercetin-*O*-hexuronide	52.93 ± 0.031 ^e^	18.18 ± 0.1 ^d^	3.76 ± 0.003 ^c^	3.4 ± 0.001 ^b^	2.42 ± 0.001 ^a^
**7**	15.8	340	447.1	285 (100)	Luteolin-*O*-hexoside	60.1 ± 0.042 ^d^	13.79 ± 0.291 ^c^	0.99 ± 0.001 ^a^	3.6 ± 0.15 ^b^	3.68 ± 0.001 ^b^
**8**	17.5	355	461.2	461 (100), 285	Luteolin-*O*-hexuronide	65.01 ± 0.001 ^e^	19.41 ± 0.154 ^a^	22.01 ± 0.004 ^b^	38.28 ± 1.342 ^d^	29.86 ± 0.001 ^c^
**9**	20.3	309	359.1	161 (100), 197 (29), 179 (18), 135	Rosmarinic acid	28.88 ± 0.001 ^c^	70.85 ± 0.001 ^d^	80.49 ± 0.001 ^e^	20.06 ± 0.32 ^a^	24.53 ± 0.037 ^b^
**10**	21.7	344	717.2	519 (100), 493 (8), 295(23)	Salvianolic acid B	nd	45.32 ± 0.106 ^b^	41.67 ± 0.001 ^a^	nd	nd
**11**	23.9	324	493.1	295 (100), 383 (6), 313 (28)	Salvianolic acid A	16.25 ± 0.001 ^a^	24.37 ± 1.848 ^c^	21.92 ± 0.001 ^b^	48.16 ± 0.037 ^e^	27.61 ± 0.001 ^d^
**12**	25	324	493.1	359 (100), 313 (8), 295	Salvianolic acid A isomer	nd	11.31 ± 0.014 ^b^	10.61 ± 0.043 ^a^	54.36 ± 0.028 ^d^	28.43 ± 0.013 ^c^
**13**	28	335	491.2	311 (100), 312 (8), 267 (4)	Salvianolic acid C	nd	7.59 ± 0.001 ^a^	nd	nd	nd
					**Total Phenolic Acids**	**45.13 ± 0.001 ^a^**	**179.73 ± 2.432 ^e^**	**170.56 ± 0.1^d^**	**122.57 ± 0.385 ^c^**	**80.57 ± 0.05 ^b^**
					**Total Isoflavonoids**	**3.46 ± 0.003 ^a^**	**-**	**-**	**-**	**-**
					**Total Flavonoids**	**185.73 ± 0.081 ^e^**	**53.43 ± 0.551 ^d^**	**27.98 ± 0.008 ^a^**	**45.88 ± 1.493 ^c^**	**36.76 ± 0.006 ^b^**
					**Total Phenolic Compounds**	**234.32 ± 0.084 ^d^**	**233.16 ± 2.984 ^d^**	**198.54 ± 0.108 ^c^**	**168.46 ± 1.878 ^b^**	**117.34 ± 0.055 ^a^**

**Table 3 antioxidants-12-00390-t003:** Overview of in vitro superoxide radical-scavenging activity and enzymatic inhibition potential of *Thymus* extracts.

Bioassay	IC_50_ Value (μg/mL)					
	T. alp	T. glb	T. pan	T. plc	T. plg	Reference Substances
**Superoxide radical inhibition**	0.91 ± 0.32 ^a^	0.95 ± 0.11 ^a^	1.21 ± 0.29 ^a^	1.11 ± 0.16 ^a^	1.12 ± 0.07 ^a^	**Gallic acid** 0.046 ± 0.001 *
***α*-Glucosidase inhibition**	366.13 ± 11.59 ^c,^*	296.82 ± 4.42 ^c,^*	388.33 ± 28.35 ^c,^*	960.29 ± 59.50 ^a^	590.44 ± 10.34 ^b^	**Acarbose** 382.18 ± 26.08 *
**AChE inhibition**	4406.38 ± 200.37 ^b^	2006.32 ± 149.81 ^a^	―	―	4962.09 ± 447.98 ^b^	**Galantamine** 3.37 ± 0.63 *

Statistical differences were assessed by one-way ANOVA, followed by Tukey’s HSD post hoc test (α = 0.05). Different lower-case letters indicate significant differences between extracts, while the presence of asterisks (*) indicates no statistical differences between reference compound and extracts. **T. alp**—*T. alpestris*, **T. glb**—*T. glabrescens*, **T. pan**—*T. pannonicus*, **T. plc**—*T. pulegioides*, **T. plg**—*T. pulcherimus*.

**Table 4 antioxidants-12-00390-t004:** Overview of in vitro MICs and MBCs of *Thymus* extracts.

Sample	*MIC/* *MBC*	*S.* *aureus*	*B.* *cereus*	*L.* *monocyt*	*S.* Typh	*E.* *coli*	*E.* *cloacae*
**T. alp**	**MIC**	1	1	1	1	1	1
	**MBC**	2	2	2	2	2	2
**T. glb**	**MIC**	1	1	1	1	1	1
	**MBC**	2	2	2	2	2	2
**T. pan**	**MIC**	1	1	2	1	1	1
	**MBC**	2	2	4	2	2	2
**T. plc**	**MIC**	1	2	2	1	1	1
	**MBC**	2	4	4	2	2	2
**T. plg**	**MIC**	2	1	2	1	1	1
	**MBC**	4	2	4	2	2	2
**Streptomycin**	**MIC**	0.1	0.025	0.15	0.1	0.1	0.025
	**MBC**	0.2	0.05	0.3	0.2	0.2	0.05
**Ampicillin**	**MIC**	0.1	0.1	0.15	0.1	0.15	0.1
	**MBC**	0.15	0.15	0.3	0.2	0.2	0.15

**T. alp**—*T. alpestris*, **T. glb**—*T. glabrescens*, **T. pan**—*T. pannonicus*, **T. plc**—*T. pulegioides*, **T. plg**—*T. pulcherimus*.

**Table 5 antioxidants-12-00390-t005:** Overview of in vitro MICs and MFCs of *Thymus* extracts.

Sample	*MIC/* *MFC*	*A. fumigatus*	*A. niger*	*A. versicolor*	*P. funiculosum*	*P. verrucosum var. cyclopium*	*T. hazarianum*
**T. alp**	**MIC**	0.5	0.5	0.25	1	0.5	1
	**MFC**	1	1	0.5	2	1	2
**T. glb**	**MIC**	0.5	0.5	0.5	1	1	0.5
	**MFC**	1	1	1	2	2	1
**T. pan**	**MIC**	0.5	0.5	0.25	0.5	0.5	0.25
	**MFC**	1	1	0.5	1	1	0.5
**T. plc**	**MIC**	0.5	0.5	0.25	1	0.5	0.5
	**MFC**	1	1	0.50	2	1	1
**T. plg**	**MIC**	0.5	1	0.5	1	1	0.5
	**MFC**	1	2	1	2	2	1
**Bifonazole**	**MIC**	0.15	0.15	0.1	0.2	0.1	0.1
	**MFC**	0.2	0.2	0.2	0.25	0.2	0.2
**Ketoconazole**	**MIC**	0.2	0.2	0.2	0.2	0.2	1
	**MFC**	0.5	0.5	0.5	0.5	0.3	1.5

**T. alp**—*T. alpestris*, **T. glb**—*T. glabrescens*, **T. pan**—*T. pannonicus*, **T. plc**—*T. pulegioides*, **T. plg**—*T. pulcherimus*.

## Data Availability

All of the data is contained within the article.

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
