# Peer review of "Thymus Species from Romanian Spontaneous Flora as Promising Source of Phenolic Secondary Metabolites with Health-Related Benefits"

_antioxidants, 2023, doi:10.3390/antiox12020390_

Round 1

Reviewer 1 Report

Dear authors, thank you very much for this interesting study. I have some comments that I think they will add a lot to your manuscript before further steps.

I think the abstract section needs more effort. For instance, in the introduction of the abstract, you mentioned all the study objectives without focusing on introducing the uniqueness of your manuscript objectives. In the methods you mentioned the wet lab work however no results were presented. And the connection between the two parts is missed in the abstract section. The conclusions are general. You should mention the novel findings related to Thymus compared to the other previous studies. Which compound proved its significant effect compared to the other groups of phytochemicals presented in the other plants?

In the introduction  section you should mention some of thymes bioactive ingredients functionalities like thymol. I suggest you to use the following reference in discussing this part that related to the impact of thymol on the functional bioactive protein structure:

https://doi.org/10.1016/j.biortech.2021.125232

 Also, in the introduction, more information about the used techniques for the extraction of the phytochemicals and the used methods for identification novelty. Additionally, why you did not perform the other spectroscopy characterization like FT-NMR and FTIR. You should focus on the objectives of your study. Introduce why your methods are important with recent references.

 2. Materials and Methods

Please clarify why you focused on maceration procedure for phytochemicals extraction? And why you just used ethanol 70% solvent? How could you ensure that the condition was optimized and the compounds were stabled? Several concerns are raised nowadays against using chemical extraction with organic solvents for obtaining phytochemicals of the mediational plants. In the paper, you mentioned that you followed alcohol. But, in your results, you did not mention any details about the comparison with other solvents. Also, have you made Single-Factor Experiments and Response Surface Methodology for extraction optimization of your used methods? You should clarify these points. Furthermore, why did you just use aerial part (e.g. flowers, leaves)? You should also clarify these parts with photos in the methods. And please add an attractive flowchart of your study that could present the importance of your plant sources.

In the results: I think the dry lab work is could add a lot to your study like molecular docking. More efforts should be for the chemical characterization presentation. I think the first section of your results needs more work to present what are the phytochemicals you found in the extract and their optimization.   Table 2: please add the unit and the signification levels. Also, how was the TF significant differences among the groups and the TEAC and DPPH should no significant differences among them?

You should clarify why we should use the phytochemicals from your source compared to the other sources. Also, you should draw figure that could clarify the mode of action that for why the different molecular composting of the different species showed different activities.

In the discussion: Generally, this section needs to be supplemented with more other literature review papers. I think you should discuss the innovations and more clarifying why your study in this section in more detail.

In conclusion: More details about your observations by showing some of this valuable study's significations point by point should be clarified. 

Author Response

Dear authors, thank you very much for this interesting study. I have some comments that I think they will add a lot to your manuscript before further steps.

I think the abstract section needs more effort. For instance, in the introduction of the abstract, you mentioned all the study objectives without focusing on introducing the uniqueness of your manuscript objectives. In the methods you mentioned the wet lab work however no results were presented. And the connection between the two parts is missed in the abstract section. The conclusions are general. You should mention the novel findings related to Thymus compared to the other previous studies. Which compound proved its significant effect compared to the other groups of phytochemicals presented in the other plants?

Thank you for your valuable recommendations! We introduced substantial changes in the content of the Abstract.

In the introduction  section you should mention some of thymes bioactive ingredients functionalities like thymol. I suggest you to use the following reference in discussing this part that related to the impact of thymol on the functional bioactive protein structure: https://doi.org/10.1016/j.biortech.2021.125232. Also, in the introduction, more information about the used techniques for the extraction of the phytochemicals and the used methods for identification novelty. Additionally, why you did not perform the other spectroscopy characterization like FT-NMR and FTIR. You should focus on the objectives of your study. Introduce why your methods are important with recent references.

Thank you for your valuable recommendations! As long as you noticed a lack of congruency and rigor in the content of the Introduction, we added supplementary informations in this section which aim to better support the background of our study. We introduced an in-depth presentation of the main bioactive constituents of thyme species which are linked with our findings presented in Results and Discussion; at the same time, a new paragraph was added in order to motivate the methodology used for extractive and analitical procedures used in our study. Of course, the use of various and complementary methods of analysis could offer an wide and detailed perspective regarding the cheamical constituents of the studied species, but, as long as the knowledge on this topic is well represented, we have chose to limitate our methodology to the most simple and suitable methods which can bring us an sufficiently detailed overview regarding qualitative and quantitative phenolic profile of our samples. Although your suggestion about FT-NMR and FTIR characterization will offer added-value to our research, at this moment we are not able to undergo this analyses as long as we are limited by time issues and the lack of expertise in this field for any of the authors.

  1. Materials and Methods

Please clarify why you focused on maceration procedure for phytochemicals extraction? And why you just used ethanol 70% solvent? How could you ensure that the condition was optimized and the compounds were stabled? Several concerns are raised nowadays against using chemical extraction with organic solvents for obtaining phytochemicals of the mediational plants. In the paper, you mentioned that you followed alcohol. But, in your results, you did not mention any details about the comparison with other solvents. Also, have you made Single-Factor Experiments and Response Surface Methodology for extraction optimization of your used methods? You should clarify these points. Furthermore, why did you just use aerial part (e.g. flowers, leaves)? You should also clarify these parts with photos in the methods. And please add an attractive flowchart of your study that could present the importance of your plant sources.

The extraction method and its parameters were chosen based on two main concerns, the traditional use of this species and pharmacopoeial regulations regarding the herbal drug extracts respectively. First of all, thyme species are cited as being traditionally used as raw herbal drugs to obtain herbal teas (infusions, using hot water as extraction solvent) or extracts with different consistencies (liquid/soft/dry extracts); in the category of liquid extracts are also classified tinctures, defined by European Pharmacopoeia 10.8 (Ph.Eur. 10.8) as „…liquid extraction preparations that are obtained using either 1 part by mass of herbal drug and 10 parts by mass or volume of extraction solvent, or 1 part by mass of herbal drug and 5 parts by mass or volume of extraction solvent… prepared by either maceration or percolation, using ethanol of a suitable concentration to extract the herbal drug, or by dissolving a soft or dry extract of the herbal drug (which has been produced using the same extraction solvent as would be used to prepare the tincture by direct extraction) in ethanol of the required concentration. ”. 70% ethanol is required for the preparation of tinctures according to Romanian Pharmacopoeis 10th edition (Tincturae monograph).

Of course, your concerning about the safety and efficacity of classic solvents and extraction procedures constitute an high interest topic at the moment, but, as we mentioned in the introduction, the use of alcohol and maceration in our study and  is based on the officially well-established references; moreover, the toxic effects of alcohol are minimized through the intermediary processing of the extracts (evaporation followed by freeze-drying). Regarding the use of optimized extraction procedures, we are totally agree with your findings (our team have a good expertise in ths area, proven by our previously-published works - https://doi.org/10.1039/D0FO02783A , https://doi.org/10.3390/antiox11061123, including optimization of extractive process of other Thymus species https://doi.org/10.1016/j.ultsonch.2022.105954 ) but, unfortunately, our present study isn’t focused to optimize the extraction procedure of the phytochemical constituents from the choosem Thymus species.

Regarding the plant material, we used the whole aerial parts of thyme species involved in this study („… aerial parts of T. alpestris …collected during flowering period”) as long as this are mentioned both by traditional and official sources (pharmacopoeias, monographs, articles) as constituent of the herbal drug Serpylli herba.

Regarding your requests about photos of the plant material, we added them separately in the graphical abstract of the revised manuscript.

In the results: I think the dry lab work is could add a lot to your study like molecular docking. More efforts should be for the chemical characterization presentation. I think the first section of your results needs more work to present what are the phytochemicals you found in the extract and their optimization.   Table 2: please add the unit and the signification levels. Also, how was the TF significant differences among the groups and the TEAC and DPPH should no significant differences among them?

Thank you for your valuable suggestions! Regarding your observation about adding data provided by „dry lab work” (molecular docking studies respectively), we weren’t focused on this type of evaluation because the bioactive potentials proven for our extracts are the result of synergistic/antagonistic effects of all of their specific constituents and their quantitative occurrence, not just from individual main constituents. At the same time, as well as for the previous comment (comment no2), none of the authors have expertise in this field. Of course, the molecular mechanisms provided by molecular docking studies for the main compounds of our extracts are already known and could help us to better explain the activity of our extracts, so, based on data available in literature, we introduced in the discussion section some information on this topic. Thank you again for this valuable recommendation!

Regarding the content of the first section of Results, we re-organized it. Please, chech in the content of manuscript. It is not clear what you meant by „needs more work to present what are the phytochemicals you found in the extract and their optimization”… can you be more explicite, please?

Table 2 was updated with requested data!

Regarding the statistic differences between the obtained data, we need to notice that not only the phenolic constituents are involved in the antioxidant potential of the extracts. TFC could be used as an orientative parameter which can offer a general overview regarding the antioxidant potential of the extracts, but it is not mandatory to have the same trend as the values provided by the antioxidant assays. At the same time, the differences are calculated between two different sets of data, so we find it normal to obtain different trends regarding statistical differences.

You should clarify why we should use the phytochemicals from your source compared to the other sources. Also, you should draw figure that could clarify the mode of action that for why the different molecular composting of the different species showed different activities.

Thank you for your recommendations! We tried to bring supplementary clarifications regarding the mentioned aspects. As well, in order to facilitate the global understanding of the manuscript, we updated our submission with a graphical abstract.

In the discussion: Generally, this section needs to be supplemented with more other literature review papers. I think you should discuss the innovations and more clarifying why your study in this section in more detail.

Thank you for your recommendations! We updated the content of this section!

In conclusion: More details about your observations by showing some of this valuable study's significations point by point should be clarified.

Thank you for your recommendations! We updated the content of this section!

Reviewer 2 Report

Manuscript proposed by Babotă and co-workers (antioxidants-2168403) entitled “Thymus species from Romanian spontaneous flora as promising source of phenolic secondary metabolites with health-related benefits presents the analysis of the phenolic profile, antioxidant and enzyme-inhibitory potential of the hydroethanolic extracts obtained from Thymus alpestris, T. glabrescens ssp. glabrescens, T. panonicus ssp. auctus, T. pulcherimus and T. pulegioides ssp. Pulegioides.
In my opinion, due to the several drawbacks, errors, and lack of description, the manuscript needs major revisions.

My major comments are presented below.

Major concerns:

- Abstract - present the importance of the aim of the work

- Introduction - what is the novelty of presented methodology and presented topic?

- Introduction – the advantages of LC-MS in the presented kind of study should be mentioned

- materials and methods, section 2.4 – pages 3 and 4 - the details of LC-MS analysis should be presented – column type, temperature of analysis, flow, injection volume, mobile phase composition, gradient conditions – did the Authors optimize the LC separation method?

- materials and methods, section 2.4 – pages 3 and 4 - the details of LC-MS analysis, including ESI-MS parameters (nebulizing gas flow, temperature, potential etc.), ion mode analysis, collision energy, should be presented – did the Authors optimize the MS and MS/MS conditions?

- table 2, page 9 – why the m/z values are presented as integers? What was the accuracy and precision of MS analysis?

- table 2, page 9 – the calculated and observed m/z values should be presented

- Both in the manuscript and supplementary data there is a lack of LC-MS data

- the quality of presented figures is low

- conclusions should present both the novelty and benefits of the presented method

- Changes in the text are needed

- Check and correct English

Author Response

Manuscript proposed by Babotă and co-workers (antioxidants-2168403) entitled “Thymus species from Romanian spontaneous flora as promising source of phenolic secondary metabolites with health-related benefits” presents the analysis of the phenolic profile, antioxidant and enzyme-inhibitory potential of the hydroethanolic extracts obtained from Thymus alpestris, T. glabrescens ssp. glabrescens, T. panonicus ssp. auctus, T. pulcherimus and T. pulegioides ssp. Pulegioides. In my opinion, due to the several drawbacks, errors, and lack of description, the manuscript needs major revisions. My major comments are presented below.

Major concerns:

- Abstract - present the importance of the aim of the work

Thank you for your valuable recommendations! We introduced substantial changes in the content of the Abstract.

- Introduction - what is the novelty of presented methodology and presented topic?the advantages of LC-MS in the presented kind of study should be mentioned

Thank you for your valuable recommendations! We added supplementary informations in this section which aim to better support the background of our study and highlight the issues pointed by you.

- materials and methods, section 2.4 – pages 3 and 4 - the details of LC-MS analysis should be presented – column type, temperature of analysis, flow, injection volume, mobile phase composition, gradient conditions – did the Authors optimize the LC separation method? pages 3 and 4 - the details of LC-MS analysis, including ESI-MS parameters (nebulizing gas flow, temperature, potential etc.), ion mode analysis, collision energy, should be presented – did the Authors optimize the MS and MS/MS conditions?

Thank you for your observations! Since we have a significant number of research papers published with the characterization of phenolic compounds, we avoided the detailed description of the methods to prevent self-plagiarism. However, we have now inserted all the requested information. Regarding the LC and MS method, the optimization was performed and published in other research paper, that is cited in the manuscript in section 2.4 (Reference 14 - Bessada, S.M.F.; Barreira, J.C.M.; Barros, L.; Ferreira, I.C.F.R.; Oliveira, M.B.P.P. Phenolic profile and antioxidant activity of Coleostephus myconis (L.) Rchb.f.: An underexploited and highly disseminated species. Ind. Crops Prod. 2016, 89, 45–51, doi:10.1016/j.indcrop.2016.04.065.).

- table 2, page 9 – why the m/z values are presented as integers? What was the accuracy and precision of MS analysis? the calculated and observed m/z values should be presented

To answer to your question we just to need to take into account that the phenolic characterization was performed using a Linear Ion Trap LTQ XL mass spectrometer (not an Orbitrap) with a resolution of m/z 100-2000. As you are aware, the nominal mass of an ion, molecule, or radical is the sum of the nominal masses of the elements in its elemental composition. So, the nominal mass of an element is the integer mass of the most abundant naturally occurring stable isotope; sometimes referred to as the principal isotope (or deprotonated ion as we referred in the manuscript). The information regarding the precision and accuracy of the MS equipment was added to the manuscript, that stated “The full scan covered the mass range from m/z 100–2000, with a mass accuracy of 0.15 Da, a peak Width of 07 FWHM, and a scan rate of 16,667 Da/sec.” Having this in mind, we have a significant degree of certainty to present the m/z integers values. However, we also understand that we have to take into account the error coupled to this, so we added one significant number to the values.

- Both in the manuscript and supplementary data there is a lack of LC-MS data

As we already mentioned in the above response, we tried to improve the inconvenients related to the lack of LC-MS data in the revised version of manuscript.

- the quality of presented figures is low

As we mentioned in the materials and methods section, the figures were generated by using GraphPad Prism 9 software. We tried to improve their quality by inserting new ones with an increased resolution.

Conclusions should present both the novelty and benefits of the presented method. Changes in the text are needed. Check and correct English

We updated the content of Conclusions section and tried to improve overall the quality of the manuscript during the revision process considering both your recommendations and those ones from pointed by Reviewer 1.

Round 2

Reviewer 1 Report

Since the authors have adjusted the manuscript according to the mentioned comments, therefore, the manuscript could be accepted in its current form.

Author Response

Dear reviewer,

We grateful for your all your valuable recommendations that helped us to improve the content of our manuscript. Thank you for your prompt response and your final decision!  

Reviewer 2 Report

In the revised version of the manuscript, the authors presented comments and answers for all of my question. However, there are still some drawbacks and mistakes:
- check the "technical" errors like presentation of Celsius degrees

- in my opinion, the low measurement accuracy in MS experiment presented by the authors in the answer, does not give unambiguous identification of the tested systems on the basis of only two signals. It should be discussed or explained.

- did the authors optimize the collision energy in MS/MS mode?

- what was the solvent used for sample preparation for HPLC and LC-MS analysis?

- did the authors observe some radical anions in the MS analysis?

Author Response

Dear reviewer, we are really sorry that our previous answer did not clarified enough your concerns. However, regarding the collision energy optimization, as we stated before, was not performed, since we used the same LC and MS conditions described in Bessada et al (2006). The solvents used were throughly described and added to the manuscript for the first round of revisions. Regarding radical anions, they were not observed in the MS analysis. Finally, the accurracy of the MS experiment is given by the equipment itself, and allows to tell you that the equipment used is one of the best equipments for screening analysis of phenolic compounds. The identification of phenolic compounds, as for many other type of compounds, using only the Full MS and MS2 signals as been performed by our research group (and hudrends of other researchers worldwide) for many years now, and peer-revieweed and publisched in many top Q1 journals in the field. The two signanls (using a good accuracy and collision energy) are enough to perform a tentative identification of the compounds, namely the flavonoids and isoflavonoids group that is only separated the sugar moitey and aglycone between Full MS and MS2. Regarding the phenolic acids identified, all rosmarinic acid derivatives, the MS2 fragments allows the identification of the type of compounds comaring to previous literature data. In the discussion section of the manuscript, it is given a thourough explanation of all the compounds, incluing the fragments and literature used for comparision.
